# Mental and substance use disorders and food insecurity among homeless adults participating in the At Home/Chez Soi study

James Lachaud [1]*, Cilia Mejia-Lancheros[1], Ri Wang[1], Kathryn Wiens[1], Rosane Nisenbaum[1,2], Vicky Stergiopoulos[1,3,4], Stephen W. Hwang[1,5], Patricia O'Campo[1,6]

1 MAP|Centre for Urban Health Solutions, Li Ka-Shing Knowledge Institute, St. Michael's Hospital, Unity Health Toronto, Toronto, ON, Canada, 2 Applied Health Research Centre, St Michael's Hospital, Li Ka Shing Knowledge Institute, Toronto, ON, Canada, 3 Centre for Addiction and Mental Health, Toronto, ON, Canada, 4 Department of Psychiatry, University of Toronto, Toronto, ON, Canada, 5 Division of General Internal Medicine, Department of Medicine, University of Toronto, Toronto, ON, Canada, 6 Dalla Lana School of Public Health, University of Toronto, Toronto, ON, Canada

* james.lachaud@unityhealth.to

**Data Availability Statement:** Data cannot be made publicly available for both ethical and legal reasons. They were collected from randomized trial

## Abstract

### Background

Few studies have examined how food insecurity changes over time when living with severe mental disorders or substance use disorders. This study identifies food insecurity trajectories of homeless adults participating in a trial of a housing intervention and examines whether receiving the intervention and having specific mental and substance disorders predict food insecurity trajectories.

### Materials and methods

We studied 520 participants in the Toronto site of the At Home/Chez-Soi project. Food insecurity data were collected at seven times during a follow-up period of up to 5.5 years. Mental and substance use disorders were assessed at baseline. Food insecurity trajectories were identified using group based-trajectory modeling. Multinomial logistic regression was used to examine the effects of the intervention and mental and substance use disorders on food insecurity trajectories.

### Results

Four food insecurity trajectories were identified: *persistently high food insecurity*, *increasing food insecurity*, *decreasing food insecurity*, and *consistently low food insecurity*. Receiving the intervention was not a predictor of membership in any specific food insecurity trajectory group. Individuals with major depressive episode, mood disorder with psychotic features, substance disorder, and co-occurring disorder (defined as having at least one alcohol or other substance use disorder and at least one non-substance related mental disorder] were more likely to remain in the persistently high food insecurity group than the consistently low food insecurity group.

implemented within a hospital setting, St. Michael's Hospital in Toronto, which conferred the participants the status of patient. Data also contain information related to mental health status of the participants. Data collection, use, and disclosure are governed by the Personal Health Information Protection Act (PHIPA, 2004) and must not be disclosed without their written informed consent, as was stated in the written informed consent form by law. As the study addresses a specific and small subpopulation, any combination of three to four variables can facilitate the identification of some participants. Nonetheless, Home/Chez Soi Toronto Data will be available to investigators for studies that have received approval from research ethics boards. Study proposals and data access requests should be sent to Evie Gogosis at Evie. Gogosis@unityhealth.to.

**Funding:** The At Home/Chez Soi research demonstration project was made possible through a financial contribution from Health Canada provided to Mental Health Commission of Canada. This study was financially supported from research grants from Ontario Ministry of Health and Long-Term Care (HSRF #259), and the Canadian Institute of Health Research (CIHR MOP-130405). Initials of authors who received Grants: HSRF #259: VS and SWH CIHR operating grant: MOP-130405: VS, PO and SWH URLs to sponsors' websites: Canadian Institute of Health Research: http://www.cihr-irsc.gc.ca/e/193.html Ontario Ministry of Health and Long-Term Care: http://www.health.gov.on.ca/en/ The funders had no role in study design, data collection and analysis, decision to publish, or preparation of the manuscript.

**Competing interests:** Authors declare no competing interests.

## Conclusion

A persistently high level of food insecurity is common among individuals with mental illness who have experienced homelessness, and the presence of certain mental health disorders increases this risk. Mental health services combined with access to resources for basic needs, and re-adaptation training are required to enhance the health and well-being of this population.

## Introduction

Mental and substance use disorders remain a major public health and social issue among homeless individuals. A recent study conducted in high-income countries found alcohol dependence and drug dependence are among the most common disorders among the homeless population, with a prevalence ranging from 8–58% and 5–54%, respectively [1]. Prevalence estimates for mental disorders, such as psychosis, depression, personality disorder, and post-traumatic stress disorder, are also higher than those reported for the general population of those countries [1,2]. For instance, the prevalence of psychosis among homeless individuals (3–42%) is approximately 3 times higher than the estimate in the general population [1].

Individuals with poor mental health are at a greater risk of poverty and job loss [3,4], social isolation/exclusion and stigmatization [5–7], which can contribute to other personal vulnerabilities [5,6,8–11], such as homelessness and food insecurity. Food insecurity is the lack of or limited access to food or to nutritious diet because of financial constraints, and is estimated to affect more than two thirds of individuals who experience homelessness with a mental disorder. [12–17] While having a severe mental disorder is an underlying contributor to food insecurity [11,18], the relationships are bidirectional; food insecurity is also a risk factor for mental health problems, including depressive symptoms, mood disorder, stress and anxiety, [19–22] suicidality, [19,23,24] substance abuse, [15,25] and poor cognitive performance [26,27].

Prior studies indicate homeless individuals disproportionately experience food insecurity more than the general population [12,13,28,29]. However limited research has quantified the effects of mental disorders on food insecurity over time among people experiencing homelessness [30,31]. Using the At Home/Chez Soi (AH/CS) data, a previous analysis across 5 Canadian cities examined the effect of a Housing First (HF) intervention on food security among homeless people with mental health disorders, and found marginal but inconsistent improvements in food security following provision of housing after 2-year of implementation [31]. In this study, mental health disorders were not the main focus, and food insecurity was modelled at specific points in time, which ignores food security trajectories altogether or assumes that they are homogenous within each intervention group. Recent studies suggest this type of analysis may conceal significant heterogeneity of long-term trajectories within groups [32,33]. Alternatively, by examining the complexity of food insecurity by modelling trajectories, additional insights into these relationships can be revealed. A trajectory-based approach can highlight how trajectories of food insecurity differ across individuals and identify factors that influence individuals to follow a given trajectory.

To build on existing literature, this paper examines the association of mental and substance use disorders with food insecurity trajectories among a sample of homeless adults enrolled in the At Home Chez Soi (AH/CS) Study, a randomized controlled trial of Housing First in Toronto. Specifically, the objectives of the study were to 1) identify trajectories of food insecurity over a period of 5.5 years, 2) test the predictive effect of the housing intervention on the

identified patterns of changes in food insecurity, and 3) analyze how mental and substance disorders predict food insecurity trajectory membership. We hypothesize that having a severe mental or substance use disorder will be strong predictors of a trajectory of persistent food insecurity over the follow up period.

## Materials and methods

### Housing First intervention

This study used data from the Toronto site of the AH/CS study, which was a randomized controlled trial that compared a scattered-site Housing First intervention and support services (HF) to treatment as usual (TAU) [34]. Detailed information on the recruitment and study design has been published elsewhere [35]. In brief, participants were recruited from community agencies, shelters, clinics, and directly from the street between October 2009 and July 2011. The inclusion criteria specified participants to be at least 18 years old, absolutely homelessness or precariously housed, diagnosed with a mental disorder, and not being served by an assertive community treatment program at the time of enrolment.

Prior to randomization, participants were stratified by their level of needs for mental health services. Need level was determined based on an algorithm that included the presence of psychotic disorder or bipolar affective disorder with psychotic symptoms (based on the Mini International Neuropsychiatric Interview (MINI) 6.0), level of community functioning (Multnomah Community Ability Scale), presence of a co-morbid substance use disorder, and history of hospitalization and incarcerations [34,35]. Out of the 575 Toronto participants, 197 participants met criteria to be classified as high level of need and 378 as moderate level of need. Participants were then randomly assigned to the intervention groups according to their level of need, with the high needs group receiving Housing First with assertive community treatment (HF-ACT) and the moderate needs group participants receiving Housing First with intensive case management services (HF-ICM). Alternatively, participants randomized to TAU continue to have access to the same locally available housing or social support services, irrespective of need level. Follow up interviews occurred every 3 months for the first two years after randomization. Once the 2-year follow-up was complete in July 2013 (phase I), the Toronto site received additional funding to follow participants until March 2017 (phase II).

### Ethical approval

The Toronto AH/CS study received approval by the Research Ethics Board of St. Michael's Hospital in Toronto, Canada. All study participants provided written informed consent to participate in the AH/CS study. The AH/CS study is registered with the International Standard Randomized Control Trial Number Register (ISRCTN42520374).

### Outcomes

Food security data were collected during in person interviews using the modified version of the US Adult Food Security Survey Module [36]. This instrument has been validated as a measure of food insecurity among individuals who experience homelessness [37–39]. The total score is a sum of 10 items related to food access, which is further dichotomized into food secure (a total score of 0 to 2) and food insecure (a total score of 3–10) [40–42]. Data were collected every 6 months during phase I and every 12 months during phase II for a total of seven time points over the follow-up period. To ensure trajectories could be properly examined, we excluded participants who had food security data for less than two time points (n = 55), resulting in a final sample of 520 participants for the analysis. Comparisons of socio-demographics

characteristics between participants and the excluded group were conducted using student t-tests or Fisher's test, and showed a statistically significant different only for the variable level of need. Out of the 520 study participants, 48.9% had a high level of need compared to 32.9% in the excluded group (p-value: 0.027, see S1 Table). Therefore, we included this variable as an adjustment variable in our models. S2 Table of Means (and standard deviation) of the number of food insecurity assessments of the participants included in this analysis by intervention group and by key mental health disorders is shown in S2 Table.

## Mental and substance use disorders

The following mental health disorders were examined as predictors of food insecurity: depressive episode, panic disorder, mood disorder with psychotic features, posttraumatic stress disorder (PTSD), manic or hypomanic episodes, psychotic disorder, alcohol and substance use disorder. All mental and substance use disorders were identified based on DSM-IV criteria using the MINI 6.0 and were evaluated at the time of screening for study eligibility [34,43].

Co-occurring disorders were defined as a comorbid condition including at least one alcohol or other substance use disorder and at least one non-substance related mental disorder. [44,45]

## Covariates

We included the factors used to screen or assign participants to the different AH/CS study groups: level of needs (high vs. moderate level of need), gender (male/female), self-identified ethnic group (white vs. member of non-white/ethnic groups), and intervention group (At home participant vs treatment as usual) [34,35,46]. Based on previous studies on food insecurity, we also adjusted for age, lifetime duration of homelessness (less than 3 years vs. 3 years or more), and education level (middle/high school, completed high school, and university or higher) [42,47].

## Statistical analyses

Group-based trajectory modelling was used to identify clusters of individuals who followed a similar pattern of change (trajectory) of food insecurity over time [48]. Using intercept and time as change parameters, the model assumed a logistic distribution of the dichotomous food insecurity variable in order to estimate the latent trajectory groups. We tested the shape of trajectory groups by including higher-order polynomial growth factors (linear, quadratic, and cubic time factor). The optimal number of trajectory groups was determined by using the Bayesian Information Criterion (BIC), while the best fit of trajectory shape was examined using the average posterior probability measure and the weighted odds of correct classification [48,49]. All models were estimated using the command Traj in Stata 15 [48].

Next, we used multinomial logistic regression to examine the effect of HF on trajectory group membership. We fit an unadjusted model including only the intervention group and an adjusted model controlling for gender, age, education level, ethno-racial status, levels of need, and lifetime duration of homelessness.

We also used multinomial logistic regression to separately assess the effect of having a mental or substance use disorder on food insecurity trajectory membership. For each mental health outcome we adjusted for gender, age, education level, ethno-racial group, levels of need, homelessness lifetime duration, and intervention group. Twenty-five individuals (4.8%) were excluded from the multivariable analysis due to missing values for either education or lifetime duration of homelessness. To evaluate the family-wise error rate due to mutilple inferences, we

used Bonferroni to compute a corrected overall critical P-value [50]. All analyses were conducted using Stata 15 [51].

## Results

Characteristics of study participants are summarized in Table 1. Out of our 520 participants, substance use disorder (37.9%), psychotic disorder (36.3%), and major depressive episode (36.0%) were the most common mental disorders identified. Moreover, 246 of the participants were diagnosed with co-occurring mental and substance use disorders (47.3%).

### Food insecurity trajectories

Based on the Bayesian Information Criterion fit statistics, the model with two quadratic trajectories and two linear trajectories was the best fit model (BIC for the 2-group model = -1635.22; $BIC_3$ = -1579.35, and $BIC_4$ = -1573.34, and $BIC_5$ = 1598.37). The average posterior probability for the 4-group model ranged from 0.75 to 0.81 and the odds of correct classification weighted posterior probability was higher than 5, which also indicate good fit.

**Table 1. Participants' characteristics at baseline.**

| Variable | Total (n = 520) |
|---|---|
| Female | 165 (31.7) |
| Age (mean(SD)) | 40.3 (11.8) |
| Self-identified ethnic group | 304 (58.5) |
| *Education level* | |
| Middle/high school | 241 (46.3) |
| Completed high school | 94 (18.1) |
| Graduate/postgraduate | 170 (32.7) |
| Missing | 15 (2.9) |
| *Level of need* | |
| High level | 175 (33.7) |
| Low level | 345 (66.3) |
| *Lifetime duration of homelessness* | |
| Less than 3 years | 225 (43.3) |
| More than 3 years | 270 (51.9) |
| Missing | 25 (4.8) |
| *Intervention* | |
| Housing first | 282 (54.2) |
| Treatment as usual | 238 (45.8) |
| *Mental and substance disorders* | |
| Major depressive episode | 187 (36.0) |
| Manic or Hypomanic episode | 57 (11.0) |
| Post-traumatic stress disorder | 126 (24.2) |
| Panic disorder | 74 (14.2) |
| Mood disorder with psychotic features | 107 (20.6) |
| Psychotic disorder | 189 (36.3) |
| Alcohol disorder | 151 (29.0) |
| Substance disorder | 197 (37.9) |
| Co-occurring disorders | 246 (47.3) |

**Table 2. Food Insecurity (FI) trajectories.**

| Parameters | | Four-group model of Food insecurity trajectory | | | |
|---|---|---|---|---|---|
| | | Intercept | linear | Quadratic | Cubic |
| | Persistent high FI | 1.12 | 0.08 | | |
| | Increasing FI | 0.08 | -0.06 | 0.00 | |
| | Decreasing FI | -0.10 | 0.16 | -0.01 | |
| | Consistent low food FI | -1.23 | -0.06 | | |
| Membership and Posterior probability of assignment | | | | | |
| | Group | Group membership: N (%) | Group APP | OCC weighted | |
| | Persistent high FI | 131 (25.2) | 0.75 | 12.74 | |
| | Increasing FI | 99 (18.9) | 0.78 | 7.70 | |
| | Decreasing FI | 163 (31.4) | 0.77 | 9.94 | |
| | Consistent low food FI | 127 (24.4) | 0.81 | 13.39 | |

APP = Average of the maximum posterior probability of assignments.

OCC = Odds of correct classification weighted posterior probability.

The model classified the participants into four food insecurity trajectory groups, *persistently high food insecurity*, *increasing food insecurity (from moderate to high)*, *decreasing food* insecurity *(from moderate to low)*, and *consistently low food insecurity* (Table 2 and Fig 1).

### Housing First and food insecurity trajectory group membership

Table 3 estimates the relative risk ratios (RRRs) of belonging to each food insecurity trajectory group (considering *consistently low FI* as the reference) for a participant assigned to the HF intervention compared to TAU. Assignment to the Housing First intervention was not statistically associated with any of food insecurity trajectory groups.

### Mental health disorders and food insecurity trajectory membership

Participants with major depressive episode were more likely to be part of the the *persistent food insecurity* trajectory group compared to the group of *consistent low food insecurity* trajectory (RRR = 1.9 [95% CI: 1.1 to 3.2]) (Table 4). Likewise, having a mood disorder with psychotic features was also associated with increased risk of being in the *persistent food insecurity* group (RRR = 3.4 [95% CI: 1.6 to 6.9]), *decreasing FI* group (RRR = 4.1 [95% CI: 2.0 to 8.1]), and *increasing FI* group (RRR = 2.7 [95% CI: 1.2 to 6.2]). Individuals with substance disorder were also more likely to be part of the group that followed the *persistent food insecurity* trajectory (RRR = 3.0 [95% CI: 1.7 to 5.2]) and the *decreasing FI* trajectory (RRR = 2.2 [95% CI:1.3 to 3.8]). In contrast, participants with psychotic disorder were less likely to belong to the *persistent food insecurity* trajectory group (RRR = 0.5 [95% CI: 0.3 to 0.9]) compared to the group of consistent low *food insecurity* trajectory. Finally, co-occurring disorders were associated with increased likelihood of being classified in the *persistent food insecurity trajectory* (RRR = 2.78 [95% CI: 1.60 to 4.83]) and in the *decreasing food insecurity trajectory* (RRR = 2.45 [95% CI: 1.45 to 4.15]) groups, compared to the group of consistent low *food insecurity* trajectory. Estimates for Mood disorder with psychotic features, Substance disorder, and Co-occurring disorders remained statistically significant after the Bonferroni adjustement for multiple inferences.

## Discussion

This study employed a group-based approach to identify trajectories of food insecurity and to examine the predictive effects of a Housing First intervention and mental and substance use

disorders on trajectory group membership. We identified four trajectory groups: 1) *persistent food insecurity* trajectory group, who remained food insecure over the entire study period, 2) *consistent low food insecurity* trajectory group, who remained low food insecure over the follow-up period, 3) *decreasing food insecurity* trajectory group, who experienced less food insecurity over time, and 4) *increasing food* insecurity trajectory group, who experienced more food insecurity over time. These trajectories highlight the diverse experiences of food insecurity among homeless individuals and a need for targeted interventions to reduce food insecurity. For example, individuals within the *increasing* trajectory group require strategies to stabilize or reverse the increasing food insecurity over time, while the *decreasing FI* insecurity group requires supports to accelerate the reductions in food insecurity.

Our findings suggest the Housing First intervention does not have a significant association with food insecurity group trajectory membership. This lack of effect is consistent with the findings of a prior analysis by O'Campo *et al.* [31] which raised concerns about the limited effects of Housing First on food insecurity over two years of follow up. In fact, the existing literature provides little evidence that the Housing First intervention improves other non-housing outcomes, such as quality of life, physical community integration, psychological community integration or mental health. [52]

Homeless individuals with major depressive episode, mood disorder with psychotic features, and substance disorder were more likely to remain persistently food insecure over the study period. We also found co-occurring disorders played a major role in food insecurity trajectories. Having an alcohol- or drug-related disorder along with a non-alcohol/drug disorder was highly associated with membership to the *persistent food insecurity* trajectory group, compared to the group of *the consistent low food insecurity* group.

These results are in line with previous studies that exposed the relationship between mental disorders and food insecurity within the homeless population[30,31,53,54]. A study conducted by Parpouchi *et al.* [30] found that among a sample of homeless individuals in Canada, those having panic disorder, PTSD or major depressive episode, or having two or more mental disorders had higher odds to be food insecure. A previous analysis conducted by O'Campo using At Home/Chez Soi two years of follow-up data found that having more psychiatric symptoms and substance use disorder were associated with lower food security. [31] In a representative sample of Canadian adults, mood or anxiety disorders were associated with severe household food insecurity, while co-occurrence of chronic physical and mental health conditions has a gradient effect on the severity of household food insecurity. [18] Another study in the United States found similar results regarding the association between depressive symptoms and persistent food insecurity among low-income rural families. [11] However, we found no prior studies that analyzed food insecurity trajectory over time.

Surprisingly, we found that having a psychotic disorder was associated with lower risk of following a *persistent food insecurity* trajectory. We found no previous studies focused on psychotic disorders and persistent food insecurity, which implies a need for more research prior to asserting that individuals with psychotic disorders are less likely to have food insecurity. It is possible that our participants experiencing psychotic disorder tended to report lower food insecurity scores due to the nature of the disorders, which can impair the perception of their every life experiences and temporality [55,56] including food-related pattern and needs.

Three plausible mechanisms may explain the relationship between mental and substance disorders and food insecurity trajectories among individuals with homelessness experience. First, mental disorders impair coping strategies and skills to manage food insufficiency in the face of scarce resources even after being in stable housing [18,57]. While previous studies identify issues related to food storage or access to a kitchen for meal preparation [23], we hypothesize that mental disorders, mainly Mood disorder with psychotic features and substance

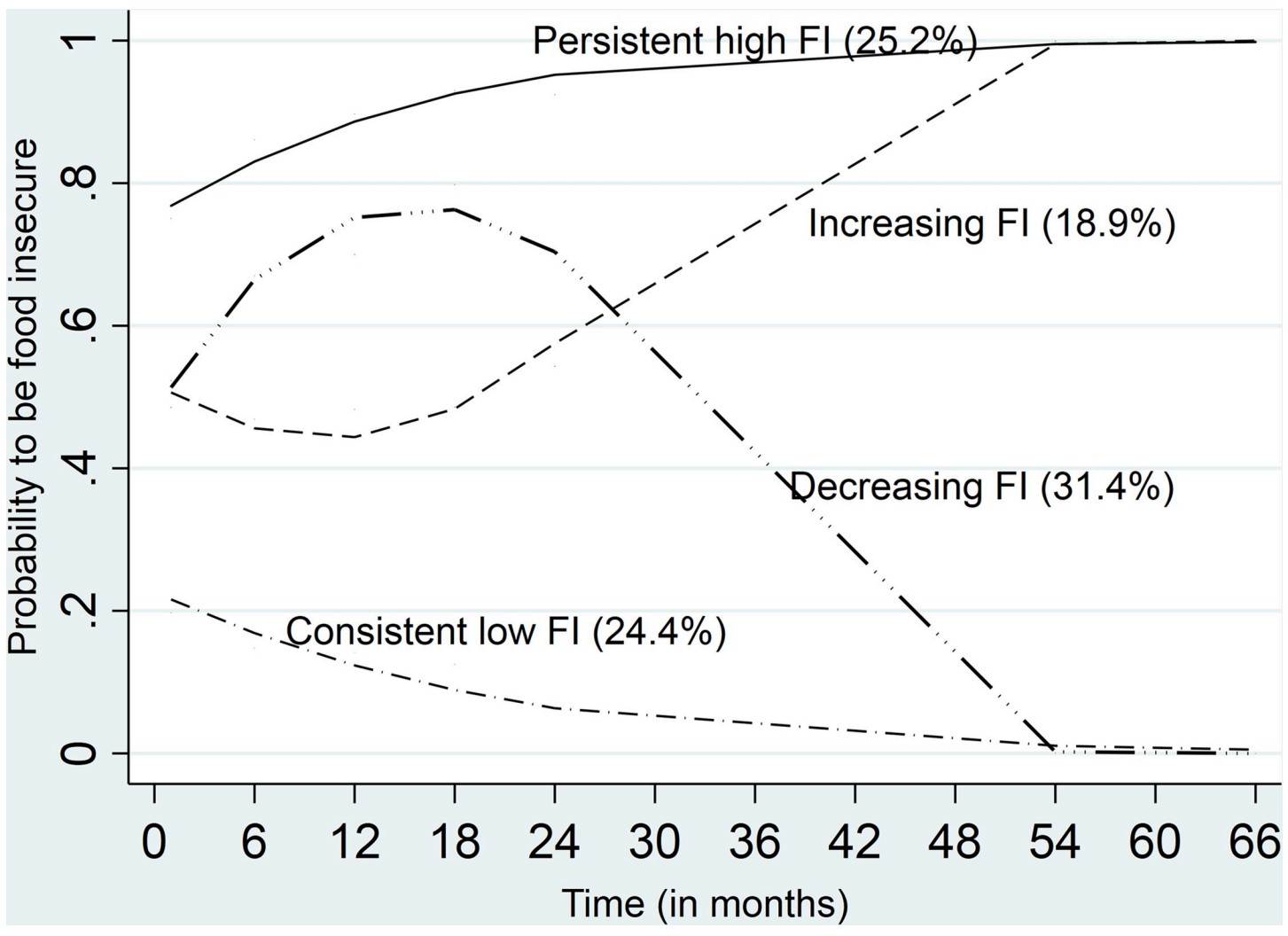

**Fig 1. Food Insecurity (FI) trajectories over time points.**

disorder, limit adjustment to am environment and lifestyle with better food security related supports (i.e., kitchen) and ability to manage resources in a manner that prioritized food security [58]. Second, when mental disorders co-occur with (recent or present) homelessness and

**Table 3. Effect of Housing First intervention on Food Insecurity (FI) trajectory group membership.**

| Intervention | Persistent FI | | Increasing FI | | Decreasing FI | |
|---|---|---|---|---|---|---|
| | RRR (95% CI) | P-value | RRR (95% CI) | P-value | RRR (95% CI) | P-value |
| *Unadjusted model* | | | | | | |
| HF vs. TAU | 0.86 (0.54 to 1.38) | 0.53 | 0.90 (0.51 to 1.60) | 0.72 | 0.97 (0.6 to 1.52) | 0.88 |
| *Adjusted model** | | | | | | |
| HF vs. TAU | 0.88 (0.54 to 1.42) | 0.59 | 0.93 (0.52 to 1.70) | 0.80 | 0.97(0.6 to 1.54) | 0.90 |

**Consistent low FI** is the base outcome

RRR: relative risk ratio

*The model is adjusted for the following variables: Gender, age, education level, self-identified ethnic group, level of need, lifetime homelessness, and intervention group

**Table 4. Multivariable multinomial logistic regressions Food Insecurity (FI) trajectories and mental disorders adjusted for baseline factors.**

| | Food Insecurity (FI) Trajectories | | | | | |
| --- | --- | --- | --- | --- | --- | --- |
| | Persistent high FI | | Increasing FI | | Decreasing FI | |
| *Mental and substance use disorder* | RRR (95% CI) | P-value | RRR (95% CI) | P-value | RRR (95% CI) | P-value |
| Major depressive episode | 1.87 (1.10 to 3.17) | 0.021 | 1.25 (0.66 to 2.36) | 0.491 | 1.31 (0.79 to 2.19) | 0.299 |
| Manic or Hypomanic episode | 1.1 (0.51 to 2.36) | 0.804 | 0.55 0.18 to 1.61) | 0.274 | 1.11 (0.54 to 2.29) | 0.772 |
| Post-traumatic stress disorder | 1.75 (0.97 to 3.14) | 0.062 | 0.9 (0.42 to 1.91) | 0.785 | 1.51 (0.85 to 2.66) | 0.156 |
| Panic disorder | 1.46 (0.70 to 3.04) | 0.316 | 1.26 (0.53 to 3.02) | 0.597 | 1.46 (0.72 to 2.94) | 0.294 |
| Mood disorder with psychotic features | 3.35 (1.63 to 6.91) | **0.001** | 2.70 (1.18 to 6.18) | 0.018 | 4.06 (2.02 to 8.15) | **0.001** |
| Psychotic disorder | 0.54 (0.31 to 0.92) | 0.024 | 0.70 (0.36 to 1.35) | 0.290 | 0.92 (0.56 to 1.53) | 0.759 |
| Alcohol disorder | 1.32 (0.75 to 2.34) | 0.334 | 1.20 (0.62 to 2.34) | 0.590 | 1.26 (0.73 to 2.18) | 0.402 |
| Substance disorder | 2.96 (1.69 to 5.21) | **0.001** | 1.36 (0.70 to 2.66) | 0.363 | 2.20 (1.28 to 3.78) | 0.004 |
| Co-occurring disorders | 2.78 (1.60 to 4.83) | **0.001** | 1.50 (0.78 to 2.86) | 0.221 | 2.45 (1.45 to 4.15) | **0.001** |

RRR: relative risk ratio

Reference group: Consistent low food FI

The model was adjusted for the following variables: Gender, age, education level, self-identified ethnic group, level of need, lifetime homelessness, and intervention group

Bonferroni Corrected overall critical P-value: 0.0014 (See the smile plot S1 Fig)

Boldface indicates statistical significance after Bonferroni Adjustment

structural discrimination, an individual's ability to gain suitable employment and social capital is impaired, which further limits availability of economic resources to support achieving food security [59–61]. Third, moving to stable housing may eventually increase living costs—transportation, energy bills, household supplies—which result in more competition for scarce resources impacting food security.

The following limitations should be considered when interpreting the present study findings. The group-based trajectories are not fixed pathways that remain invariant over time or across populations, but rather help to identify clusters of people following similar pathways in relation to a given outcome. [48,49] Thus, more research studies are needed to verify whether the trajectories are similar in other homeless populations, such as individuals without severe mental illness. Moreover, participants were asked at each interview about food insecurity over the past month. These recall data may be subject to bias and error. Moreover, personality disorder was not included among the mental disorders screened by the At Home/Chez Soi project, though it is generally highly prevalent among homeless individuals [1]. Therefore, we could not include in the present study.

Our findings have research and public health implications. Since mental health issues and food insecurity are often intertwined chronic conditions, [11,18] it is necessary to account for the unique impact specific mental disorders can have on food insecurity over a long period of time. This is especially relevant when providing housing, as individual mental health needs and trauma should be taken into account but also to ensure access to adequate social and basic needs, including food security program. Homelessness, food insecurity and discrimination are avoidable social problems that affect thousands of people locally and globally; political and social actions such as greater access to social and transitional housing, skill and job support services, a basic income for low income individuals, and interventions to target implicit bias for employers and service providers can help to create healthier and more inclusive societies.

## Conclusion

Homeless individuals with mental health disorders can be grouped into food insecurity trajectories, and those with mental health disorders such as major depressive episode, Mood disorder with psychotic features, and substance disorder are more likely to remain persistently food insecure over time. Mental health services combined with access to resources for basic needs, and social and re-adaptation training are required to enhance the health and well-being of this population.

## Supporting information

**S1 Fig. Smile plot of relative risk ratio by food security trajectory groups.**
(DOCX)

**S1 Table. Fisher test comparing characteristics between study participants (n = 520) and the excluded group (n = 55).**
(DOCX)

**S2 Table. Means (and standard deviation) of the number of food insecurity assessments of the participants included in this analysis by intervention group and by key mental health disorders.**
(DOCX)

**S3 Table. Participants' characteristics at baseline by intervention group.**
(DOCX)

**S4 Table. Participants' characteristics at baseline by food security trajectory group.**
(DOCX)

## Acknowledgments

We thank the At Home/Chez Soi participants whose willingness to share their lives, experiences and stories with us made this project possible. We also thank the At Home/Chez Soi project team, site coordinators and service providers who have contributed to the design, implementation and follow-up of participants at the Toronto site.

## Author Contributions

**Conceptualization:** James Lachaud, Cilia Mejia-Lancheros, Patricia O'Campo.

**Data curation:** Ri Wang.

**Formal analysis:** James Lachaud.

**Funding acquisition:** Vicky Stergiopoulos, Stephen W. Hwang, Patricia O'Campo.

**Investigation:** James Lachaud, Cilia Mejia-Lancheros, Patricia O'Campo.

**Methodology:** James Lachaud, Cilia Mejia-Lancheros, Rosane Nisenbaum.

**Project administration:** Patricia O'Campo.

**Validation:** James Lachaud, Ri Wang, Kathryn Wiens, Rosane Nisenbaum, Vicky Stergiopoulos, Stephen W. Hwang, Patricia O'Campo.

**Writing – original draft:** James Lachaud.

**Writing – review & editing:** James Lachaud, Cilia Mejia-Lancheros, Ri Wang, Kathryn Wiens, Rosane Nisenbaum, Vicky Stergiopoulos, Stephen W. Hwang, Patricia O'Campo.

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
