## [Decision Letter · Decision Letter 0]

7 Oct 2019

PONE-D-19-21014

Mental and Substance Use Disorders and Food Insecurity among Homeless Adults Participating in the At Home/Chez Soi Study

PLOS ONE

Dear Dr Lachaud,

Thank you for submitting your manuscript to PLOS ONE. After careful consideration, we feel that it has merit but does not fully meet PLOS ONE’s publication criteria as it currently stands. Therefore, we invite you to submit a revised version of the manuscript that addresses the points raised during the review process.

The manuscript has been assessed by two reviewers; their comments are available below.

The reviewers have raised major concerns that need attention in a revision. The reviewers note that a related paper from the same project should be discussed and the relationship between the current work and that publication described in greater detail. The reviewers request additional information on the participants excluded, and raise questions about the statistical analyses undertaken, including the need for correction for multiple comparisons for some of the analyses. The reviewers request further information on the restrictions that apply to data access and note that causal language should be revised as causal conclusions are not supported by the study design.

Could you please carefully revise the manuscript to address the concerns raised by the reviewers?

We would appreciate receiving your revised manuscript by Nov 19 2019 11:59PM. Please include the following items when submitting your revised manuscript:

We look forward to receiving your revised manuscript.

Kind regards,

Iratxe Puebla

Senior Managing Editor, PLOS ONE

Journal Requirements:

Reviewers' comments:

Reviewer's Responses to Questions

**Comments to the Author**

1. Is the manuscript technically sound, and do the data support the conclusions?

Reviewer #1: Partly

Reviewer #2: Partly

2. Has the statistical analysis been performed appropriately and rigorously? 

Reviewer #1: Yes

Reviewer #2: Yes

3. Have the authors made all data underlying the findings in their manuscript fully available?

Reviewer #1: No

Reviewer #2: No

4. Is the manuscript presented in an intelligible fashion and written in standard English?

Reviewer #1: Yes

Reviewer #2: Yes

5. Review Comments to the Author

Reviewer #1: Although the statistical analyses were conducted appropriately and rigorously to the best of my knowledge, the following major concerns prompted me to question the technical soundness of the manuscript:

The manuscript cites “a recent study” on food insecurity among homeless individuals (lines 84-85), but does not make it clear in the text that this study is an analysis of data from the same parent study (At Home/Chez Soi) as the current manuscript. Because the research questions of the published 2017 paper on food insecurity among At Home/Chez Soi participants appear related to the research questions of the current manuscript, the authors should acknowledge the previously published paper and explicitly state how the research questions and analyses of the current paper are distinct.

The manuscript repeatedly uses the language of impact with reference to substance use disorders and mental disorders, e.g. “this paper examines the impact of mental and substance use disorders on food insecurity trajectories” (lines 93-94). This does not accurately reflect the paper’s research design and analyses, which demonstrate associations between substance use/mental disorders and food insecurity trajectories, but cannot definitively show that substance use/mental disorders impact food insecurity.

The plausible mechanisms suggested (lines 294-306) speak mainly to the general relationship between homelessness and food insecurity (already well-established in prior research) and do not match the study's specific findings on food insecurity trajectories. For example, the authors mention lack of food storage and pressures of survival needs (e.g. finding a place to sleep) as explanatory factors in high rates of food insecurity among homeless people, but there was no difference in food insecurity noted between Housing First and treatment as usual participants in this study. Presumably the Housing First residents would have greater access to food storage, and are not faced with the survival need of finding a place to sleep, so the relevance of this mechanism to the study's findings is not clear.

The statement that “political and social actions are required to create healthier and more inclusive societies” (lines 321-322) is extremely vague; it would be more helpful for the authors to name specific policy or practice implications related to the study findings. The conclusion “Both food related and mental health interventions and services are required to enhance the health and well being of this population” (lines 326-328) is also very vague and does not speak to the study’s findings regarding different food insecurity trajectories and the possibility that people with different vulnerabilities are at varying levels of risk for food insecurity and therefore may require different services or interventions.

Regarding data availability, the authors state that “Data cannot be shared publicly because of ethical restrictions to the data” but do not specify the nature of the restrictions or further elaborate on this.

In addition, I noted the following minor concerns:

Use of the term “alcohol and drug dependence” in line 65 is outdated, as this reflects DSM-IV language.

The origin of the definition of food insecurity provided (lines 74-75) is not clear; many current definitions specify that food insecurity is not just about access to food, but access to nutritious food. It is also not clear if the stated prevalence of food insecurity in homeless individuals with mental disorders is referring to the Canadian context, or globally (line 76).

The meaning of “ethno-racial group (yes or no)” is not clear (line 150).

Gender is not listed as a covariate on p. 7-8, but is named as a control variable in Tables 3 and 4.

The authors discuss dichotomizing the variable of food insecurity (lines 135-136), but then reference three levels of food insecurity (low, moderate, and high) in defining the four food insecurity trajectories (lines 189-191).

Reviewer #2: The authors present results of a study of food insecurity over time in homeless individuals in Toronto participating in a randomized trial of a scattered-site Housing First intervention with support services compared to treatment as usual. The authors identify four subgroups of trajectories of food insecurity (persistently high food insecurity, increasing food insecurity, decreasing food insecurity, and consistently low food insecurity). They found no association between intervention group and trajectory group, but did identify some mental health disorders at higher risk of belonging to certain food insecurity groups. The manuscript will be strengthened if the authors consider the following points.

1. The authors mention that 55 individuals were excluded due to having only 1 food insecurity assessment. This makes sense in the context of the focus on longitudinal patterns in food insecurity. However, I think it will help the reader understand the sample better if the authors include some basic descriptions of those excluded (how many from HF vs TAU, how many were food insecure vs not, how many were high vs moderate level of need).

2. What was the mean (and standard deviation) of the number of food insecurity assessments for the 520 people included in this analysis? Did this differ by intervention group or by key mental health disorders of interest?

3. It would be helpful to show Table 1 also broken down by Intervention group. It also would be helpful to have the distribution of need (moderate vs high) by mental and substance disorders presented in a table or in the text (which might help in interpreting the results in later tables).

3. Are the quadratic terms really needed for the Increasing FI and Decreasing FI groups? The parameters presented in Table 2 are essentially 0. What is the BIC in the 4 group model with all linear trajectories?

4. Under the section starting on line 216, it would be worth reporting the percent randomized to HR in each of the FI trajectory groups to further support the RRR presented in Table 3.

5. In some sense, the trajectory groups might be considered ordered, with Persistent FI the worst, then Increasing FI, then Decreasing FI, and then consistently low FI. Why did the authors use a multinomial logistic model rather than an approach that takes into account this ordering? Taking this into account might better capture who is most at risk for the worse trajectories compared to less worse trajectories.

6. The authors perform 9 different models in Table 4, but there is no adjustment of multiple comparisons.

7. For co-occurring disorders, the authors focus on the results related to persistent FI (lines 239-242), but there was also a finding with increased risk fo decreasing FI.

8. How do the authors interpret the fact that substance abuse and mood disorder with psychotic features are at increased risk of both the persistent FI (a bad thing) and decreasing FI (probably a good indicator)? Same for co-occurring disorders. These conclusions may be different if the authors actually take into account the ordering of the groups as I mention above, because as analyzed, these results are confusing to try to interpret.

Minor edits:

1. line 65: change "vary" to "varies"

2. line 140: there is an extra "."

3. line 150: there is an extra ","

4. Table 1: It would be helpful to know what was captured in Ethno-racial group. There are 304 of these individuals in the sample, so I imagine there are further sub-groups that can be listed here with corresponding frequencies to give the reader a better sense of who is represented in this sample.

5. Table 2: the percentage for Increasing FI should be 19.0 (not 18.9)

6. line 237: what does "and RRR=1.6 [95% CI: 1.0 to 2.7]" refer to? This doesn't seem to match anything in Table 4.

7. line 258 "need to for" should be "need for"

8. In the Abstract, Discussion , and Conclusion, the authors include PTSD in the list of disorders more likely to be in the persistently FI group, but this finding was not significant (though close). The authors also don't mention this in the text of the Results section.

6. PLOS authors have the option to publish the peer review history of their article (what does this mean?). If published, this will include your full peer review and any attached files.

Reviewer #1: No

Reviewer #2: No

---

## [Author Response · Author response to Decision Letter 0]

19 Nov 2019

PONE-D-19-21014

Mental and Substance Use Disorders and Food Insecurity among Homeless Adults Participating in the At Home/Chez Soi Study

Editorial Team

Dear Editor, 

I am pleased to re-submit the manuscript entitled “Mental and Substance Use Disorders and Food Insecurity among Homeless Adults Participating in the At Home/Chez Soi Study”. We would like to extend our sincere gratitude to the PlosOne team and the Reviewers for their positive and constructive feedback. 

The co-authors and I have carefully reviewed the Reviewers’ comments and provide point-by-point responses to their comments below. In particular, we have addressed the concerns related to the participants excluded, and have also applied the Bonferroni test for multiple comparisons for some of the analyses. We have also changed the statement to the data restrictions and removed all language or expressions that can imply causal relationship. 

We include a manuscript with the highlighted changes and a clean version, as well as an appendix file with additional tables. We believe that the revised manuscript is greatly improved and has addressed all Reviewer comments. 

Thank you again for your consideration.

Sincerely,

Reviewer #1: Although the statistical analyses were conducted appropriately and rigorously to the best of my knowledge, the following major concerns prompted me to question the technical soundness of the manuscript:

The manuscript cites “a recent study” on food insecurity among homeless individuals (lines 84-85), but does not make it clear in the text that this study is an analysis of data from the same parent study (At Home/Chez Soi) as the current manuscript. Because the research questions of the published 2017 paper on food insecurity among At Home/Chez Soi participants appear related to the research questions of the current manuscript, the authors should acknowledge the previously published paper and explicitly state how the research questions and analyses of the current paper are distinct.

Response: We thank Reviewer #1 for this constructive feedback and appreciate this important point raised. We added more information about this study and explained how is distinct to this manuscript. We state as follows (lines 84-90):

Using the At Home/Chez Soi (AH/CS) data, a recent study across 5 Canadian cities examined the effect of a Housing First (HF) intervention on food security among homeless with mental health disorders, and found marginal but inconsistent improvements in food security following provision of housing after 2-year of implementation(31). In this study, mental health disorders were not the main focus, and food insecurity was modelled at specific points in time, which ignores food security trajectories altogether or assumes that they are homogenous within each intervention group. Recent studies suggest this type of analysis may conceal significant heterogeneity of long-term trajectories within groups (32,33).

In the following paragraph, we explained (line 96-104):

To build on existing literature, this paper examines the association of mental and substance use disorders on food insecurity trajectories among a sample of homeless adults enrolled in the At Home Chez Soi (AH/CS) Study, a randomized controlled trial of Housing First in Toronto. Specifically, the objectives of the study were to 1) identify trajectories of food insecurity over a period of 5.5 years, 2) test the predictive effect of the housing intervention on the identified patterns of changes in food insecurity, and 3) analyze how mental and substance disorders predict food insecurity trajectory membership. 

Reviewer: The manuscript repeatedly uses the language of impact with reference to substance use disorders and mental disorders, e.g. “this paper examines the impact of mental and substance use disorders on food insecurity trajectories” (lines 93-94). This does not accurately reflect the paper’s research design and analyses, which demonstrate associations between substance use/mental disorders and food insecurity trajectories, but cannot definitively show that substance use/mental disorders impact food insecurity.

Response: We changed as follows (lines 96):

To build on existing literature, this paper examines the association of mental and substance use disorders on food insecurity trajectories among a sample of homeless adults enrolled in the At Home Chez Soi (AH/CS) Study, a randomized controlled trial of Housing First in Toronto.

Reviewer: The plausible mechanisms suggested (lines 294-306) speak mainly to the general relationship between homelessness and food insecurity (already well-established in prior research) and do not match the study's specific findings on food insecurity trajectories. For example, the authors mention lack of food storage and pressures of survival needs (e.g. finding a place to sleep) as explanatory factors in high rates of food insecurity among homeless people, but there was no difference in food insecurity noted between Housing First and treatment as usual participants in this study. Presumably the Housing First residents would have greater access to food storage, and are not faced with the survival need of finding a place to sleep, so the relevance of this mechanism to the study's findings is not clear.

Response: We have reformulated the entire paragraph in the discussion section (lines 316-327): 

First, mental disorders impair coping strategies and skills to manage food insufficiency in the face of scarce resources even after being in stable housing(18,56). While previous studies identify issues related to food storage or access to a kitchen for meal preparation (23), we hypothesize that mental disorders, mainly Mood disorder with psychotic features and substance disorder, limit adjustment to an environment and lifestyle with better food security related supports (i.e., kitchen) and ability to manage resources in a manner that prioritized food security(57). Second, when mental disorders co-occur with (recent or present) homelessness and structural discrimination, an individual’s ability to gain suitable employment and social capital is impaired, which further limits availability of economic resources to support achieving food security(58–60). Third, moving to stable housing may eventually increase living costs—transportation, energy bills, household supplies--which result in more competition for scarce resources impacting food security. 

Reviewer: The statement that “political and social actions are required to create healthier and more inclusive societies” (lines 321-322) is extremely vague; it would be more helpful for the authors to name specific policy or practice implications related to the study findings. The conclusion “Both food related and mental health interventions and services are required to enhance the health and wellbeing of this population” (lines 326-328) is also very vague and does not speak to the study’s findings regarding different food insecurity trajectories and the possibility that people with different vulnerabilities are at varying levels of risk for food insecurity and therefore may require different services or interventions.

Response: We reformulated the entire paragraph as follows to provide more specification (lines 336-345): 

Since mental health issues and food insecurity are often intertwined chronic conditions,(11,18) it is necessary to account for the unique impact specific mental disorders can have on food insecurity over a long period of time. This is especially relevant when providing housing, as individual mental health needs and trauma should be taken into account but also to ensure access to adequate social and basic needs, including food security program. Homelessness, food insecurity and discrimination are avoidable social problems that affect thousands of people locally and globally; political and social actions such as greater access to social and transitional housing, skill and job support services, a basic income for low income individuals, and interventions to target implicit bias for employers and service providers can help to create healthier and more inclusive societies. 

Mental health services combined with access to resources for basic needs, and social and re-adaptation training are required to enhance the health and well-being of this population. (lines 350-352)

Reviewer: Regarding data availability, the authors state that “Data cannot be shared publicly because of ethical restrictions to the data” but do not specify the nature of the restrictions or further elaborate on this.

Response: We changed the statement as follows

Anonymised participant data, the study protocol, informed consent forms, survey forms, and statistical analysis plan from the At Home/Chez Soi Toronto site study will be available to investigators for studies that have received approval from independent research committees or research ethics boards. Data are available from the publication date of this article onwards. Study proposals and data access requests should be sent to Dr Stephen Hwang at Stephen.Hwang@unityhealth.to . All study proposals and data requests will be further reviewed by the At Home/Chez Soi team at the Toronto site. Data sharing agreements between the requestors and At Home/Chez Soi principal investigators need to be completed before accessing the data.

Reviewer: In addition, I noted the following minor concerns:

Use of the term “alcohol and drug dependence” in line 65 is outdated, as this reflects DSM-IV language.

Response: Thank you for the precision. We reformulated it as follows (line 65):

While the prevalence of mental disorders vary across studies, alcohol and drug use are the most common disorders among the homeless population, with prevalence ranging from 8-58% and 5-54%, respectively(1)

We don’t used “alcohol and drug disorders” to avoid too much repetition of the work “disorder” in the same sentence

Reviewer: The origin of the definition of food insecurity provided (lines 74-75) is not clear; many current definitions specify that food insecurity is not just about access to food, but access to nutritious food. It is also not clear if the stated prevalence of food insecurity in homeless individuals with mental disorders is referring to the Canadian context, or globally (line 76).

We have added “or to nutritious diet” to the definition (line 75)

The meaning of “ethno-racial group (yes or no)” is not clear (line 150).

We have added it: (defined as white vs. any ethno-racial or no-white groups) (line 163). 

This self-identified ethno-racial was included in Toronto Housing First project not to identify specific (race or ethnicity), but to adapt the services the category. 

Goering PN, Streiner DL, Adair C, Aubry T, Barker J, Distasio J, et al. The at Home/Chez Soi trial protocol: A pragmatic, multi-site, randomised controlled trial of a Housing First intervention for homeless individuals with mental illness in five Canadian cities. BMJ Open. 2011;1(2):1–18.

Gender is not listed as a covariate on p. 7-8, but is named as a control variable in Tables 3 and 4.

We have it as a covariate: gender (male/female) line (162)

Reviewer: The authors discuss dichotomizing the variable of food insecurity (lines 135-136), but then reference three levels of food insecurity (low, moderate, and high) in defining the four food insecurity trajectories (lines 189-191).

Thank you. We explained in the methodology how we transformed the original variable presented in the lines 135-136 into the food insecurity trajectory groups (lines 169-177)

We sincerely appreciate the constructive feedback from Reviewer #1! 

Reviewer #2: The authors present results of a study of food insecurity over time in homeless individuals in Toronto participating in a randomized trial of a scattered-site Housing First intervention with support services compared to treatment as usual. The authors identify four subgroups of trajectories of food insecurity (persistently high food insecurity, increasing food insecurity, decreasing food insecurity, and consistently low food insecurity). They found no association between intervention group and trajectory group, but did identify some mental health disorders at higher risk of belonging to certain food insecurity groups. The manuscript will be strengthened if the authors consider the following points.

We thank Reviewer #2 for these insightful suggestions and recommendations. 

1. Reviewer: The authors mention that 55 individuals were excluded due to having only 1 food insecurity assessment. This makes sense in the context of the focus on longitudinal patterns in food insecurity. However, I think it will help the reader understand the sample better if the authors include some basic descriptions of those excluded (how many from HF vs TAU, how many were food insecure vs not, how many were high vs moderate level of need).

Response: We have added a table A1 in Appendix comparing participant’s socio-demographic characteristics collected at the baseline, including age, gender, Intervention group, level of need, ethno-racial group. Let us mention the exclusion is for those who had less than 2 assessments, including no food data at all.

We also indicated in the text (line 14-5148):

Comparisons of socio-demographics characteristics between participants and the excluded group were conducted using student t-tests or Fisher’s test were conducted and showed statistically significant differences only for the variable level of need (See Table A1 in Appendix), which is included as an adjustment variable in our models. 

2. Reviewer: What was the mean (and standard deviation) of the number of food insecurity assessments for the 520 people included in this analysis? Did this differ by intervention group or by key mental health disorders of interest?

Response: We added a Table A2 in Appendix presenting the mean (and standard deviation by intervention groups and key mental health. 

We also indicated in the text (lines 148-151): 

Table A2 of Means (and standard deviation) of the number of food insecurity assessments of the participants included in this analysis by intervention group and by key mental health disorders is shown in Appendix. 

3. Reviewer: It would be helpful to show Table 1 also broken down by Intervention group. It also would be helpful to have the distribution of need (moderate vs high) by mental and substance disorders presented in a table or in the text (which might help in interpreting the results in later tables).

Response: We have added this comparison table in Appendix Table A3. We do not want to present in the text body since this comparison table has been published in previous papers of AH /Chez-Soi Study (so it could be seen a duplication of published results) and the comparison is not the main purpose of this study. 

The last one was recently published at the Lancet Psychiatry. 

Stergiopoulos, V., Mejia-Lancheros, C., Nisenbaum, R., Wang, R., Lachaud, J, O'Campo, P., and Hwang, S.W. (2019). The long-term effects of rent supplements and mental health support services on housing and health outcomes of homeless adults with mental illness: outcomes of the extended At Home/Chez Soi randomized controlled trial, The Lancet Psychiatry., https://doi.org/10.1016/S2215-0366(19)30371-2 

3. Reviewer: Are the quadratic terms really needed for the Increasing FI and Decreasing FI groups? The parameters presented in Table 2 are essentially 0. What is the BIC in the 4 group model with all linear trajectories?

Response: We have tested all the possibilities, including 4-group model with all linear trajectories. The BIC for 4 group model with all linear trajectories is -1632.59. In addition, the Average of the maximum posterior probability of assignment, other good-fitness criteria, is better for the model selected. 

4. Reviewer: Under the section starting on line 216, it would be worth reporting the percent randomized to HR in each of the FI trajectory groups to further support the RRR presented in Table 3.

Response: We have added Table A4 in Appendix which shows Participants’ Characteristics at baseline by Food Security Trajectory Group including the variable intervention and all key mental health to see the percent randomized. 

5. Reviewer: In some sense, the trajectory groups might be considered ordered, with Persistent FI the worst, then Increasing FI, then Decreasing FI, and then consistently low FI. Why did the authors use a multinomial logistic model rather than an approach that takes into account this ordering? Taking this into account might better capture who is most at risk for the worse trajectories compared to less worse trajectories.

Response: As you mentioned “in some sense”, we also have difficult to accept a strict order when looking at the Trajectory group graph mainly for groups 2 and 3. Hence, that is why we believe that imposing that order might bias the results and the interpretation. 

6. Reviewer: The authors perform 9 different models in Table 4, but there is no adjustment of multiple comparisons.

Response: We want to thank Reviewer#2 for this insight comment. We computed the Bonferroni Adjustment for multiple testing as suggested. We have added in the methodology: 

To evaluate the family-wise error rate due to multiple inferences, we use Bonferroni to compute a Corrected overall critical P-value (49). (lines 187-188)

The value overall critical p-value is presented at the foot of the Table 4, and the smile plot Fig A1 in Appendix. (lines 267-268)

We also explained in the section of Results (lines 257-259): 

Estimates for Mood disorder with psychotic features, Substance disorder, and Co-occurring disorders remained statistically significant after the Bonferroni adjustement for multiple inferences.

7. Reviewer: For co-occurring disorders, the authors focus on the results related to persistent FI (lines 239-242), but there was also a finding with increased risk for decreasing FI.

Response: We have added it the analysis. (lines 256-257)

8. Reviewer: How do the authors interpret the fact that substance abuse and mood disorder with psychotic features are at increased risk of both the persistent FI (a bad thing) and decreasing FI (probably a good indicator)? Same for co-occurring disorders. These conclusions may be different if the authors actually take into account the ordering of the groups as I mention above, because as analyzed, these results are confusing to try to interpret.

Response: The reference category for the analysis is Consistent low food FI, as we added as a footnote to the Table 4 (line 265). Such as, it is a better status compared to the other food insecurity trajectory groups, including Decreasing FI group. Hence, it seems that the estimates for substance abuse and mood disorder with psychotic (and also co-occurring disorders) go to the expected direction, higher than 1, compared to the Consistent low food FI group. Yet again, the fact the estimates for Increasing FI are all not statistically significant expresses our difficulty to impose a strict order between groups 2 and 3. 

Reviewer: Minor edits:

1. line 65: change "vary" to "varies"

We have changed it

2. line 140: there is an extra "."

We have removed it

3. line 150: there is an extra ","

We have removed it 

4. Table 1: It would be helpful to know what was captured in Ethno-racial group. There are 304 of these individuals in the sample, so I imagine there are further sub-groups that can be listed here with corresponding frequencies to give the reader a better sense of who is represented in this sample.

Response: We have added the definition Ethno-racial (defined as: white vs. any ethno-racial or no-white groups). (line 163)

This self-identified ethno-racial was included in Toronto Housing First project not to identify specific (race or ethnicity), but to adapt the services the category. 

Goering PN, Streiner DL, Adair C, Aubry T, Barker J, Distasio J, et al. The at Home/Chez Soi trial protocol: A pragmatic, multi-site, randomised controlled trial of a Housing First intervention for homeless individuals with mental illness in five Canadian cities. BMJ Open. 2011;1(2):1–18.

5. Table 2: the percentage for Increasing FI should be 19.0 (not 18.9)

Thank you. We have corrected it. 

6. line 237: what does "and RRR=1.6 [95% CI: 1.0 to 2.7]" refer to? This doesn't seem to match anything in Table 4.

Thanks. We have removed it

7. line 258 "need to for" should be "need for"

We have corrected it.

8. In the Abstract, Discussion, and Conclusion, the authors include PTSD in the list of disorders more likely to be in the persistently FI group, but this finding was not significant (though close). The authors also don't mention this in the text of the Results section.

We removed PTSD in this list of disorders in the Abstract, Discussion, and Conclusion. 

We thank Reviewer #2 again for all of these extremely helpful comments!

---

## [Decision Letter · Decision Letter 1]

4 Feb 2020

PONE-D-19-21014R1

Mental and Substance Use Disorders and Food Insecurity among Homeless Adults Participating in the At Home/Chez Soi Study

PLOS ONE

Dear Dr Lachaud,

Thank you for submitting your manuscript to PLOS ONE. After careful consideration, we feel that it has merit but does not fully meet PLOS ONE’s publication criteria as it currently stands. Therefore, we invite you to submit a revised version of the manuscript that addresses the points raised during the review process.

The manuscript has been evaluated by two reviewers, and their comments are available below.

The reviewers have raised a number of minor concerns that need attention. They request further clarification on the terminology used, and suggest some grammar corrections. 

Could you please revise the manuscript to carefully address the concerns raised?

We would appreciate receiving your revised manuscript by Mar 19 2020 11:59PM. To enhance the reproducibility of your results, we recommend that if applicable you deposit your laboratory protocols in protocols.io, where a protocol can be assigned its own identifier (DOI) such that it can be cited independently in the future. For instructions see: http://journals.plos.org/plosone/s/submission-guidelines#loc-laboratory-protocols

We look forward to receiving your revised manuscript.

Kind regards,

Carmen Melatti, PhD

Associate Editor

PLOS ONE

Reviewers' comments:

Reviewer's Responses to Questions

**Comments to the Author**

1. If the authors have adequately addressed your comments raised in a previous round of review and you feel that this manuscript is now acceptable for publication, you may indicate that here to bypass the “Comments to the Author” section, enter your conflict of interest statement in the “Confidential to Editor” section, and submit your "Accept" recommendation.

Reviewer #1: (No Response)

Reviewer #2: (No Response)

2. Is the manuscript technically sound, and do the data support the conclusions?

Reviewer #1: Yes

Reviewer #2: Yes

3. Has the statistical analysis been performed appropriately and rigorously? 

Reviewer #1: Yes

Reviewer #2: Yes

4. Have the authors made all data underlying the findings in their manuscript fully available?

Reviewer #1: Yes

Reviewer #2: Yes

5. Is the manuscript presented in an intelligible fashion and written in standard English?

Reviewer #1: No

Reviewer #2: Yes

6. Review Comments to the Author

Reviewer #1: The authors have addressed most of my concerns noted in the previous review. I have noted a few additional or remaining concerns:

In lines 75-77, it is still not clear if the prevalence statistic “ and is estimated to affect more than two thirds of individuals who experience homelessness with a mental disorder” is in reference to Canada, or elsewhere.

The paper is inconsistent in its terminology with regard to if “mental disorders” includes substance use disorders. The first paragraph of the introduction (starting with line 64) implies that mental disorders does include substance use disorders, but elsewhere throughout the paper (including in the title), the authors use the term “mental and substance use disorders” – suggesting that mental disorders is not inclusive of substance use disorders. The language should be edited to ensure that it is consistent throughout the paper.

In line 146, the authors note a statistically significant difference between study participants and participants excluded from the study dataset. The authors should describe what direction this difference is in, e.g. which group shows a greater level of need?

Line 97 should say “with food insecurity trajectories” instead of “on food insecurity trajectories” – this is a minor change but important in terms of not implying causality.

There are grammatical errors throughout the paper, including the word “people” missing after “homeless” in line 85. The sentences in lines 145-151 are confusing to read, due to grammatical errors (e.g. the first sentence uses the phrase “were conducted” twice). The paper should be thoroughly edited to correct such errors.

It is confusing to use the generic term “ethno-racial group” in Table 1. If this refers to people who do not identify as white, I would rephrase it here as something like “member of non-white racial or ethnic group.” In their response to reviewers, the authors write “This self-identified ethno-racial was included in Toronto Housing First project not to identify specific (race or ethnicity), but to adapt the services the category” – I don’t understand what this means. I would also recommend saying “non-white” instead of “no-white” in line 163.

In line 284, the authors should reiterate that the “prior study by O’Campo et al.” also used the At Home/Chez Soi Trial data. It is not that surprising that two analyses of data from the same study found similar findings about the effects of Housing First on food insecurity. Similarly, line 299 should be rewritten to convey that “another study conducted by O’Campo” is not an independent study, but another analysis of the At Home data.

Reviewer #2: The authors have addressed the majority of my earlier concerns. In their response to the reviews (and within the manuscript), authors refer to an Appendix with tables and a figure, but I do not see the Appendix with the submission.

There are also a couple of typographical errors:

line 44: "ressources"

line 117: "symtpoms"

7. PLOS authors have the option to publish the peer review history of their article (what does this mean?). If published, this will include your full peer review and any attached files.

Reviewer #1: No

Reviewer #2: No

---

## [Author Response · Author response to Decision Letter 1]

5 Feb 2020

Comments to the Author

Reviewer #1: The authors have addressed most of my concerns noted in the previous review. I have noted a few additional or remaining concerns:

 Comment: In lines 75-77, it is still not clear if the prevalence statistic “and is estimated to affect more than two thirds of individuals who experience homelessness with a mental disorder” is in reference to Canada, or elsewhere.

Answer: Thank you for asking for this precision. We reformulated it as follows:

A recent study conducted in high-income countries found alcohol dependence and drug dependence are among the most common disorders among the homeless population, with a prevalence ranging from 8-58% and 5-54%, respectively(1). Prevalence estimates for mental disorders, such as psychosis, depression, personality disorder, and post-traumatic stress disorder, are also higher than those reported for the general population of those countries(1,2). For instance, the prevalence of psychosis among homeless individuals (3-42%) is approximately 3 times higher than the estimate in the general population(1).

Comment: The paper is inconsistent in its terminology with regard to if “mental disorders” includes substance use disorders. The first paragraph of the introduction (starting with line 64) implies that mental disorders does include substance use disorders, but elsewhere throughout the paper (including in the title), the authors use the term “mental and substance use disorders” – suggesting that mental disorders is not inclusive of substance use disorders. The language should be edited to ensure that it is consistent throughout the paper.

Answer: The text has been edited to avoid this confusion. 

Mental and substance use disorders remain a major public health and social issue among homeless individuals. A recent study conducted in high-income countries found alcohol dependence and drug dependence are among the most common disorders among the homeless population, with a prevalence ranging from 8-58% and 5-54%, respectively(1). Prevalence estimates for other mental disorders, such as psychosis, depression, personality disorder, and post-traumatic stress disorder, are also higher than those reported for the general population of those countries(1,2). For instance, the prevalence of psychosis among homeless individuals (3-42%) is approximately 3 times higher than the estimate in the general population(1).

Comment: In line 146, the authors note a statistically significant difference between study participants and participants excluded from the study dataset. The authors should describe what direction this difference is in, e.g. which group shows a greater level of need?

Answer: Corrected as follows:

Comparisons of socio-demographics characteristics between participants and the excluded group were conducted using student t-tests or Fisher’s test, and showed a statistically significant different only for the variable level of need. Out of the 520 study participants, 48.9% had a high level of need compared to 32.9% in the excluded group (p-value: 0.027, see Table A1 in Appendix). Therefore, we included this variable as an adjustment variable in our models.

Comment: Line 97 should say “with food insecurity trajectories” instead of “on food insecurity trajectories” – this is a minor change but important in terms of not implying causality. 

Answer: Corrected

Comment: There are grammatical errors throughout the paper, including the word “people” missing after “homeless” in line 85. The sentences in lines 145-151 are confusing to read, due to grammatical errors (e.g. the first sentence uses the phrase “were conducted” twice). The paper should be thoroughly edited to correct such errors.

It is confusing to use the generic term “ethno-racial group” in Table 1. If this refers to people who do not identify as white, I would rephrase it here as something like “member of non-white racial or ethnic group.” In their response to reviewers, the authors write “This self-identified ethno-racial was included in Toronto Housing First project not to identify specific (race or ethnicity), but to adapt the services the category” – I don’t understand what this means. I would also recommend saying “non-white” instead of “no-white” in line 163.

Answer: line 85 corrected. 

lines 145-151: Thank you. We reviewed it and removed the words “were conducted”, which were typed twice. We also reviewed the entire manuscript to be sure such typos errors it doesn’t repeat.

Thank you for the suggestion for ethno-racial group. We rephrased as follows: 

“Self-identified ethnic group (white vs. member of non-white/ethnic groups)”

We also changed it in the table 1 and the footnotes of the tables 3 and 4. 

Comment: In line 284, the authors should reiterate that the “prior study by O’Campo et al.” also used the At Home/Chez Soi Trial data. It is not that surprising that two analyses of data from the same study found similar findings about the effects of Housing First on food insecurity. Similarly, line 299 should be rewritten to convey that “another study conducted by O’Campo” is not an independent study, but another analysis of the At Home data.

Answer: line 284 corrected. “prior study by..” was replaced by “prior analysis by…”

 Line 289 “another study conducted by O’Campo” was replaced by “A previous analysis conducted by O’Campo using At Home/Chez Soi two years of follow-up data found that”

Thank for your insightful comments!

Reviewer #2: The authors have addressed the majority of my earlier concerns. In their response to the reviews (and within the manuscript), authors refer to an Appendix with tables and a figure, but I do not see the Appendix with the submission.

Answer: added it directly at the end of the manuscript

Comment: There are also a couple of typographical errors:

line 44: "ressources"

Answer: Corrected

Comment: line 117: "symtpoms"

Answer: Corrected

Thank for !

---

## [Editor Report · Decision Letter 2]

7 Apr 2020

Mental and Substance Use Disorders and Food Insecurity among Homeless Adults Participating in the At Home/Chez Soi Study

PONE-D-19-21014R2

Dear Dr. Lachaud,

We are pleased to inform you that your manuscript has been judged scientifically suitable for publication and will be formally accepted for publication once it complies with all outstanding technical requirements.

With kind regards,

Markos Tesfaye, M.D., Ph.D

Academic Editor

PLOS ONE

Additional Editor Comments (optional):

Thank you for addressing all of the latest comments from the reviewers.

I see a couple of typos in the document that you might wish to correct.

1. Line 168: "non-white racial or ethnic group" probably needs to be removed.

2. (Appendix) Tables A1 and A4 : you might wish to replace "ethno-racial group as in table A3.
---

## [Editor Report · Acceptance letter]

9 Apr 2020

PONE-D-19-21014R2 

Mental and Substance Use Disorders and Food Insecurity among Homeless Adults Participating in the At Home/Chez Soi Study 

Dear Dr. Lachaud:

I am pleased to inform you that your manuscript has been deemed suitable for publication in PLOS ONE. Congratulations! Your manuscript is now with our production department. 

With kind regards,

on behalf of

Prof. Markos Tesfaye 

Academic Editor

PLOS ONE